# Nucleotide binding is the critical regulator of ABCG2 conformational transitions

**Zsuzsanna Gyöngy[1,2], Gábor Mocsár[1], Éva Hegedűs[1], Thomas Stockner[3], Zsuzsanna Ritter[1,2], László Homolya[4], Anita Schamberger[4], Tamás I Orbán[4], Judit Remenyik[5], Gergely Szakacs[4,6], Katalin Goda[1]***

[1]Department of Biophysics and Cell Biology, Faculty of Medicine, University of Debrecen, Debrecen, Hungary; [2]Doctoral School of Molecular Cell and Immune Biology, University of Debrecen, Debrecen, Hungary; [3]Institute of Pharmacology, Center for Physiology and Pharmacology, Medical University of Vienna, Vienna, Austria; [4]Institute of Enzymology, Research Centre for Natural Sciences, Budapest, Hungary; [5]Institute of Food Technology, Faculty of Agricultural and Food Sciences and Environmental Management, University of Debrecen, Debrecen, Hungary; [6]Institute of Cancer Research, Medical University of Vienna, Vienna, Austria

**Abstract** ABCG2 is an exporter-type ABC protein that can expel numerous chemically unrelated xeno- and endobiotics from cells. When expressed in tumor cells or tumor stem cells, ABCG2 confers multidrug resistance, contributing to the failure of chemotherapy. Molecular details orchestrating substrate translocation and ATP hydrolysis remain elusive. Here, we present methods to concomitantly investigate substrate and nucleotide binding by ABCG2 in cells. Using the conformation-sensitive antibody 5D3, we show that the switch from the inward-facing (IF) to the outward-facing (OF) conformation of ABCG2 is induced by nucleotide binding. IF-OF transition is facilitated by substrates, and hindered by the inhibitor Ko143. Direct measurements of 5D3 and substrate binding to ABCG2 indicate that the high-to-low affinity switch of the drug binding site coincides with the transition from the IF to the OF conformation. Low substrate binding persists in the post-hydrolysis state, supporting that dissociation of the ATP hydrolysis products is required to reset the high substrate affinity IF conformation of ABCG2.

**\*For correspondence:**
goda@med.unideb.hu

**Competing interest:** The authors declare that no competing interests exist.

## Editor's evaluation

The ABC transporter ABCG2 extrudes chemotherapy reagents and other xenobiotics from a number of different tissues. How ABCG2 operates at the molecular level has been largely derived from structures and dynamics carried out in non-physiological environments. The paper presents convincing cell-based evidence describing the relationship between structural changes of ABCG2 and substrate binding using flow cytometry, confocal microscopy, and fluorescence-correlation spectroscopy methods. Both the mechanistic conclusions and methodology employed offer important insights, which will be of general interest to the biochemistry and transport biology communities.

## Introduction

The human ABCG2 protein (also known as Breast Cancer Resistance Protein [BCRP], Mitoxantrone Resistance Protein [MXR], and Placenta-Specific ABC transporter [ABCP]) is a primary active transporter that belongs to the ATP-binding cassette (ABC) protein superfamily (*Doyle et al., 1998*).

ABCG2 can expel a large variety of chemically unrelated molecules, mainly hydrophobic or amphipathic compounds from cells, including chemotherapeutic drugs and toxic metabolic side-products such as pheophorbide A (*Jonker et al., 2002*), estrone-3-sulfate (E3S) (*Suzuki et al., 2003*), and uric acid (*Woodward et al., 2009*). ABCG2 is expressed in tissues with barrier functions including the blood-brain barrier, blood-testis barrier, intestine, liver, placenta, kidneys, and mammary glands (*Doyle et al., 1998*; *Fetsch et al., 2006*; *van Herwaarden and Schinkel, 2006*). Because of its tissue localization and its broad substrate spectrum, ABCG2 plays an important role in the absorption, distribution, elimination, and toxicity (ADME-Tox) of chemotherapeutic drugs used for the treatment of various diseases (*Giacomini et al., 2013*; *Sarkadi et al., 2006*). In addition, the expression of ABCG2 in tumor tissues, especially in the so-called tumor stem cells or drug-tolerant persister cells, correlates with an unfavorable prognosis of tumor chemotherapy (*Ding et al., 2010*). On the other hand, genetic polymorphisms leading to decreased expression and/or impaired function of ABCG2 may result in various pathological conditions (recently reviewed in *Homolya, 2021*). These include altered therapy responses, drug-related toxic reactions, as well as hyperuricemia and gout (*Ishikawa et al., 2013*), since ABCG2 is also a key player of the intestinal uric acid elimination pathway (*Woodward et al., 2009*; *Köttgen and Köttgen, 2021*).

ABCG2 is a 'half-transporter', consisting of an N-terminal nucleotide-binding domain (NBD), followed by a C-terminal transmembrane domain (TMD) that form homodimers in the plasma membrane to obtain a functional transporter (*Kage et al., 2002*). Although the substrate specificity of ABCG2 partially overlaps with that of other drug transporting ABC proteins, sequence similarity is limited to the evolutionarily conserved NBDs. In all ABC transporters, ATP is bound by the Walker A and B motifs of one NBD and by the signature sequence of the other NBD, resulting in a close interaction of the two NBDs, which is often referred to as 'sandwich-dimer' (*Locher, 2009*). The molecular mechanism of action of ABC transporters has intrigued scientists for many years. Recent publications of high-resolution cryogenic electron microscopy (cryo-EM) structures have allowed the interrogation of molecular details with atomic precision. According to the widely accepted idea, uphill transport of substrates is linked to conformational changes of the TMDs, which are regulated by the ATP-dependent formation and separation of the NBD 'sandwich-dimer.' In the absence of ATP, *apo*-ABCG2 adopts an inward-facing (IF) state, wherein the transmembrane cavity is open toward the cytosolic side, with the central four transmembrane helices (TH2, TH5, TH2', and TH5') forming a large drug-binding pocket (*cavity 1*) accessible for substrate binding (*Figure 1A*). ATP-bound ABCG2 is in an outward-facing (OF) state, allowing the release of substrates to the extracellular side (*Figure 1B*). The IF structure contains two substrate-binding cavities: a large central cavity at the cytoplasmic side (*cavity 1*) and an additional cavity (*cavity 2*) located toward the extracellular part of ABCG2 (*Figure 1A*). The two cavities are separated by L554 and L555, which form a tight hydrophobic seal at the top of *cavity 1*, preventing the entry of substrates to *cavity 2* in the IF conformation (*Taylor et al., 2017*; *Khunweeraphong et al., 2017*). In other ABCG2 IF structures captured in the presence of substrates or inhibitors, *cavity 1* is occupied by a substrate molecule or either one or two inhibitor molecules (*Jackson et al., 2018*; *Orlando and Liao, 2020*). The ATP-free IF state can be detected by the conformation-sensitive monoclonal antibody 5D3 (*Taylor et al., 2017*), which recognizes two composite epitopes formed by extracellular loops (*Figure 1C and E*) that are disengaged in the OF state (*Figure 1D and F*). The structure of the OF conformer could only be captured by using the E211Q catalytic glutamate mutant ABCG2 variant. In the ATP-bound conformer of this mutant protein, *cavity 1* is completely collapsed, the di-leucine gate is closed, while *cavity 2* is open to the extracellular side of the plasma membrane (*Manolaridis et al., 2018*). However, mutation of the catalytic glutamate may affect the energetics of conformational changes in ABC transporters; thus, it may be argued that the OF state of the E211Q mutant ABCG2 observed in the presence of ATP does not represent a true catalytic intermediate (*Manolaridis et al., 2018*; *Lusvarghi et al., 2021*).

Despite the plethora of new ABCG2 structures, several long-standing questions concerning the intramolecular crosstalk between the substrate binding and the nucleotide-binding sites remain unanswered. For example, the intramolecular mechanism by which transported substrates stimulate the ATPase activity has not been identified (*Ozvegy et al., 2001*; *Sarkadi et al., 1992*; *Telbisz et al., 2012*). Similarly, there is no universal agreement on how conformational changes induced by nucleotide binding and hydrolysis result in the decrease of drug binding affinity and lead to the release of the transported substrate. The prevailing view is that ATP binding leads to the formation

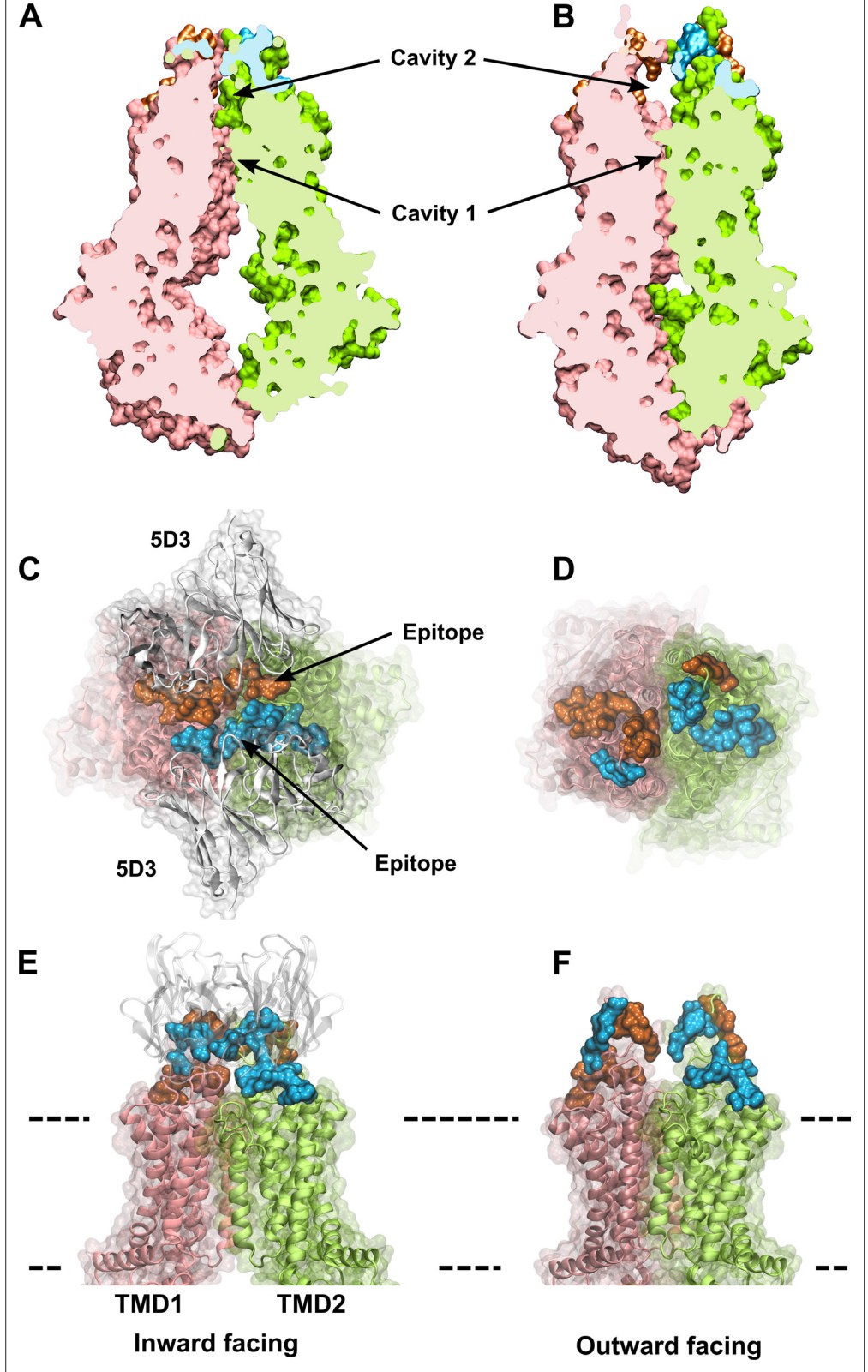

**Figure 1.** The IF and OF conformers of ABCG2. A slice through IF (**A**) and OF (**B**) ABCG2 shows the lower and upper cavity and reveal their volume changes induced by the transition. Two rotationally symmetric 5D3 epitopes are formed in the IF conformation (**C, E**) and are absent in the OF conformation of ABCG2 (**D, F**). Panels (**C**) and (**D**) show an extracellular view of the 5D3-bound IF (PDB ID: 6ETI), and the OF (PDB ID: 6HBU) ABCG2

*Figure 1 continued on next page*

*Figure 1 continued*

conformations, respectively. A side view of the IF (**E**) and the OF (**F**) ABCG2 is also shown. The two protomers of ABCG2 are colored in pink and green in cartoon backbone and transparent surface rendering. The two epitopes are highlighted in blue and orange, and include all residues that are in direct contact with the 5D3 Fab (white) in the 6ETI cryo-EM structure. The membrane boundaries are indicated by dashed lines. cryo-EM, cryogenic electron microscopy; IF, inward-facing; OF, outward-facing.

of the closed NBD 'sandwich-dimer,' switching the TMDs from the IF to the OF state, resulting in a concomitant decrease of substrate affinity (*McDevitt et al., 2008*). In that scenario, the role of ATP hydrolysis is to reset the transporter to the high substrate affinity IF state by initiating the dissociation of the closed NBD dimer (*Khunweeraphong et al., 2019*). However, an alternative model states that nucleotide binding is not sufficient, and the IF-OF transition requires ATP hydrolysis (*Kapoor et al., 2018*).

Since ABC transporters are very sensitive to the composition of their plasma membrane environment (*Januliene and Moeller, 2020*; *Pál et al., 2007*; *Bordignon et al., 2020*; *Romsicki and Sharom, 1998*), here we use fluorescence-based methods to study intramolecular crosstalk in live cells or semi-permeabilized cells, where ABCG2 molecules are present in their quasi-natural environment, in the undisturbed context of the plasma membrane (*Bársony et al., 2016*; *Goda et al., 2020*). To address how changes of TMD conformation and substrate binding are coupled to ATP binding and hydrolysis, we quantify binding of the conformation-sensitive 5D3 antibody (*Ozvegy-Laczka et al., 2005*). To characterize drug binding, we introduce confocal microscopy- and fluorescence correlation spectroscopy (FCS)-based assays. By measuring substrate binding and reactivity to 5D3, we show that nucleotide binding drives ABCG2 from a high to a low substrate-affinity state and find that this switch coincides with the flip from the IF to the OF conformation.

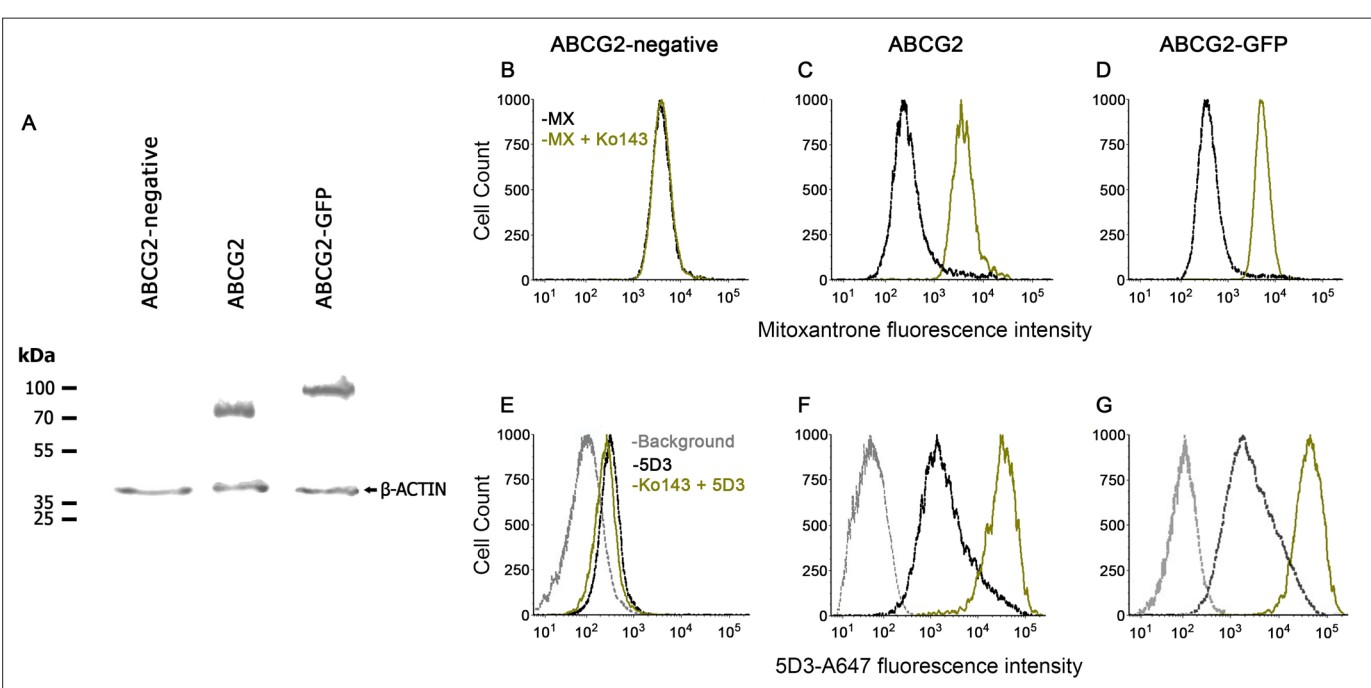

**Figure 2.** Functional characterization of human ABCG2 and ABCG2-GFP. Western blot showing comparable expression of ABCG2 and ABCG2-GFP in MDCK II cells using the BXP-21 anti-ABCG2 mAb (**A**). ABCG2 and ABCG2-GFP expressing cells show decreased mitoxantrone (MX) accumulation, which is reversed to the level of the ABCG2-negative cells by 2 μM Ko143 (**B–D**). 2 μM Ko143 pre-treatment causes an increase in the 5D3-reactivity of ABCG2 (**F**) and ABCG2-GFP (**G**) expressing cells. ABCG2-negative cells show only a negligible 5D3 binding that is not affected by Ko143 (**E**).

The online version of this article includes the following source data for figure 2:

**Source data 1.** Source Data to *Figure 2A*.

## Results

### MDCK II cells express fully functional ABCG2 and ABCG2-GFP

ABCG2 and its N-terminally GFP-tagged variant (ABCG2-GFP) were expressed at comparable levels in MDCK II cells (*Figure 2A*). In accordance with literature data (*Orbán et al., 2008*; *Haider et al., 2011*), the transport activity of ABCG2 was not influenced by the GFP-tag (*Figure 2B–D*). Previous observations showed that ABCG2 inhibitors, such as Ko143, can enhance 5D3 mAb binding by shifting the equilibrium to the IF state (*Ozvegy-Laczka et al., 2005*; *Telbisz et al., 2012*). Accordingly, in the plasma membrane of untreated live cells, ABCG2-GFP and ABCG2 exhibited comparably low 5D3-re-activity, which was increased in a similar extent by Ko143 treatment (*Figure 2E–G*), supporting the notion that the GFP-tag does not modify the conformational response of ABCG2.

### Nucleotide binding is sufficient to trigger the switch from the 5D3-reactive IF conformation to a 5D3-dim OF conformation

To study the nucleotide-dependent conformation changes of ABCG2, we systematically changed the intracellular nucleotide concentrations in semi-permeabilized cells. In accordance with an ATP-regulated switch of the TMD conformation, increasing $ATP/Mg^{2+}$ concentrations gradually decreased the 5D3-A647-reactivity of ABCG2-positive cells, with practically zero staining at high $ATP/Mg^{2+}$ concentrations (*Figure 3A and B*). To prevent nucleotide hydrolysis, ATP was either added in the absence of $Mg^{2+}$ (ATP+EDTA; *Figure 3C and D*), on ice (*Figure 3E and F*), or ATP was replaced with the non-hydrolyzable ATP analog AMP-PNP (*Figure 3G and H*). Interestingly, the conformational change driving ABCG2 into a 5D3-dim (i.e., 5D3 non-reactive) state also occurred in the absence of ATP hydrolysis and showed similar nucleotide concentration dependence (see *Table 1*). These results indicate that the 5D3-dim and 5D3-reactive conformations correspond to the OF and IF conformations as observed in ATP-bound and nucleotide-free crystal structures, respectively (*Figure 1*).

By replacing the cleaved gamma phosphate following ATP hydrolysis, phosphate analogs, such as vanadate (Vi) or beryllium fluoride (BeFx), trap ABC transporters in a stable ternary complex (ABCG2-ADP-Vi/BeFx). Based on different geometries of myosin structures obtained with transition state analogs, the BeFx- or Vi-trapped post-hydrolytic complexes are believed to represent pre- and post-hydrolytic conformations, respectively (*Smith and Rayment, 1996*, *Fisher et al., 1995*; *Szakács et al., 2000*). Co-treatment with Vi or BeFx increased the apparent nucleotide affinity of ABCG2 about 10-fold (*Table 1*) in conditions permitting ATP hydrolysis (*Figure 3A and B*), confirming that both phosphate analogs can form stable ADP-trapped complexes with ABCG2. When ATP hydrolysis was prevented, the phosphate analogs did not have any effect on the $K_A$ values (*Figure 3C–H* and *Table 1*).

Similar to the bacterial ABC transporter MsbA (*Moeller et al., 2015*), $ADP/Mg^{2+}$ could also induce the IF to OF switch, albeit at slightly higher concentrations ($K_A = 7.38 \pm 2.31$ mM) compared to $ATP/Mg^{2+}$ (see also *Table 1*). Moreover, nucleotide trapping occurred in the presence of phosphate analogs and $ADP/Mg^{2+}$ (*Figure 3I–J*). The $K_A$ values obtained with $ADP/Mg^{2+}$ and Vi ($K_A = 0.44 \pm 0.1$ mM) or BeFx ($K_A = 0.57 \pm 0.32$ mM) did not differ from the $K_A$ values of trapping reactions starting from $ATP/Mg^{2+}$. However, in the trapping reactions with $ADP/Mg^{2+}$, about 30% of ABCG2 molecules remained in a 5D3-reactive state even in the presence of very high nucleotide concentrations. This observation may suggest that the ternary complex resulting from $ADP/Mg^{2+}$ is different (i.e., less stable, possessing a shorter lifetime compared to the complex produced from $ATP/Mg^{2+}$ in the hydrolytic cycle) and therefore the trapping reaction occurs with lower efficiency. Since in energized cells the cytosolic ATP concentration is more than 10-fold higher compared to ADP concentrations (*Williams et al., 1993*), these results indicate that in live cells, (i) the switch from the 5D3-reactive IF to a 5D3-dim OF conformation is induced by ATP binding; and (ii) resetting to the 5D3-reactive IF conformation can only occur after the release of the hydrolysis products.

### Substrates increase the rate of formation of the Vi- or BeFx-trapped species

Transported substrates increase the turnover rate of ATP hydrolysis in many ABC transporters including ABCB1 and ABCG2 (*Sarkadi et al., 1992*; *Telbisz et al., 2012*). Progressive accumulation of the transporter molecules in the stable Vi- or BeF$_x$-trapped post-hydrolytic states represents a partial

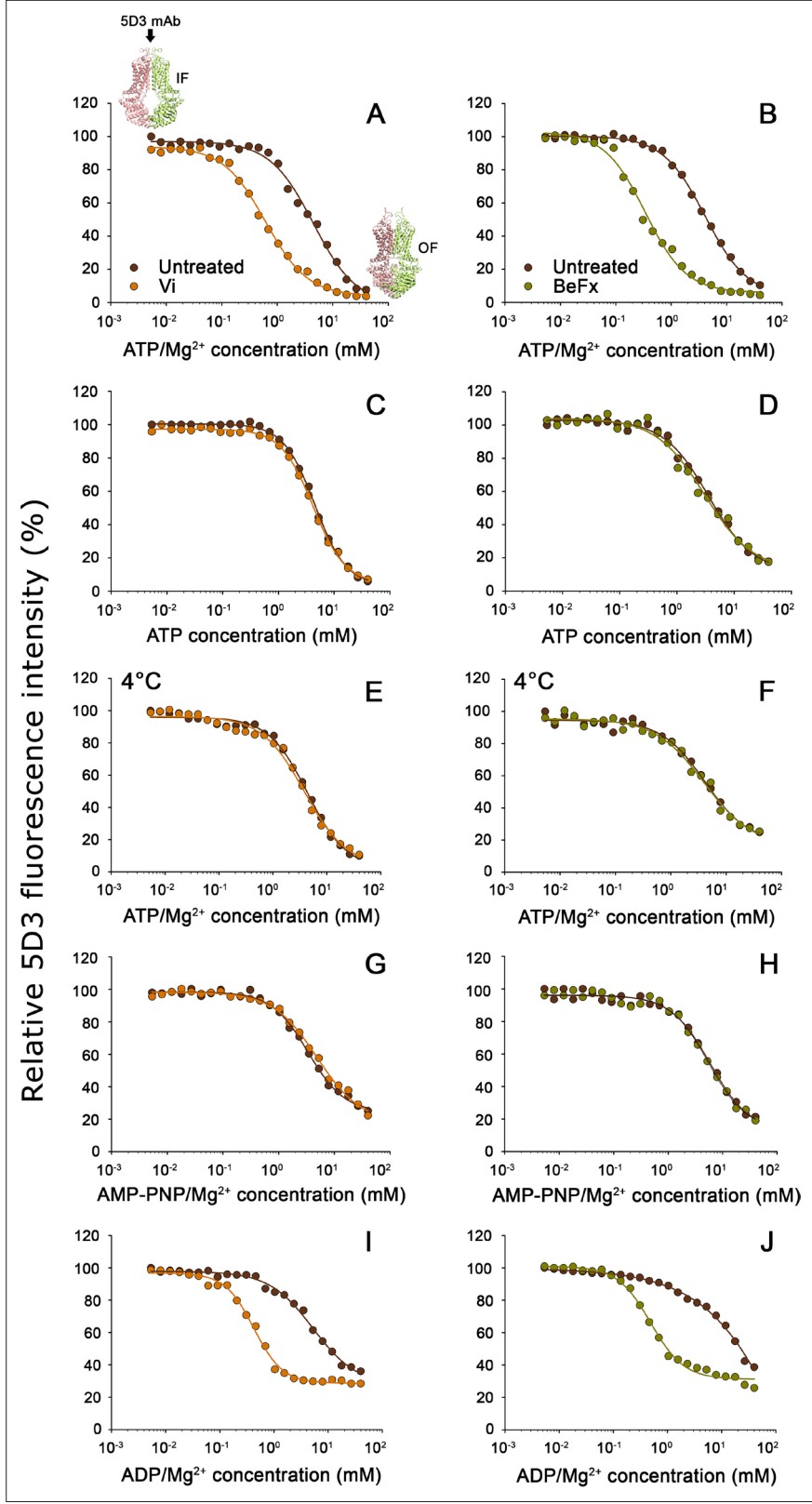

**Figure 3.** Nucleotide dependence of 5D3-reactivity. MDCK-ABCG2 cells were permeabilized to allow the titration of intracellular nucleotide concentrations. Dose-response curves of 5D3 binding with increasing concentrations of ATP/Mg$^{2+}$ (panels (**A**), and (**B**)), ATP in the presence of EDTA (panels (**C**) and (**D**)), AMP-PNP/Mg$^{2+}$ (panels (**G**) and (**H**)), or ADP/Mg$^{2+}$ (panels (**I**) and (**J**)) were obtained in the presence or absence of Vi (*left* panels) or BeFx (*right*

*Figure 3 continued on next page*

*Figure 3 continued*

panels). Samples were pre-treated with nucleotides at 37°C for 10 min, then further incubated with 5 µg/ml 5D3-A647 at 37°C for 20 min except for (panels (**E**) and (**F**)), where all the treatments were carried out on ice. In case of nucleotide trapping, permeabilized cells were co-treated with nucleotides and BeFx or Vi at 37°C for 30 min, then un-trapped nucleotides were washed out and 5D3 labeling was carried out on ice for 45 min. Representative curves are shown from three to five independent experiments. The small inserts in panel (**A**) depict the IF and OF conformations of ABCG2. IF, inward-facing; OF, outward-facing.

reaction of the catalytic cycle in ABC transporters (*Szabó et al., 1998*). Accordingly, the accumulation of ABCG2 in the Vi- or BeFx-trapped post-hydrolytic complex was accelerated by substrates (*Figure 4*). E3S and quercetin induced an about a fivefold increase in the rate of the trapping reaction (for $t_{1/2}$ values, see *Figure 4B*), which is consistent with the extent of stimulations achieved by these compounds in the ATPase assay (*Telbisz et al., 2012*).

The relatively long $t_{1/2}$ values compared to the total cycle time, which is in the order of 100 ms (as inferred from ATPase data *Yu et al., 2021*), suggest that formation of a stable post-hydrolysis complex by phosphate mimicking anions is a low-probability event. The increased rate of the trapping reaction in the presence of substrates may be explained by a higher turnover of the ATPase cycle, or a longer duration of the Vi- or BeFx-"sensitive" state, that is between the dissociation of the cleaved phosphate and the disassembly of the NBD dimer. Consistently, trapping reactions starting from ADP/$Mg^{2+}$ were not accelerated by quercetin or E3S (*Figure 4B*), suggesting that substrates do not affect the overall stability or lifespan of the ADP-bound, phosphate analog-sensitive ABCG2 conformer. This can happen, if substrates similarly accelerate the formation and the dissociation of the ADP-bound conformer, or alternatively, they do not have any effect on these processes.

## Substrates accelerate the IF to OF transition of ABCG2

With the aim to pinpoint the transition that is accelerated by transported substrates, in the following experiments, we studied how nucleotides and substrates affect the kinetics of the IF to OF transition detected by a shift in 5D3 binding. To align ABCG2 molecules in an IF state, semi-permeabilized

**Table 1.** Apparent nucleotide affinities ($K_A$) determined in 5D3-reactivity experiments.

| Treatment | Conditions | | $K_A$ (mM ± SD) | n | Statistical comparisons to ATP/$Mg^{2+}$ (1st row) | Figure |
| | Temp. | $Mg^{2+}$ | | | | |
|---|---|---|---|---|---|---|
| ATP | 37°C | + | 4.15 ± 1.08 | 11 | – | *Figure 3A* |
| ATP+Vi | 37°C | + | 0.34 ± 0.25 | 11 | p < 0.001 | *Figure 3A* |
| ATP+BeFx | 37°C | + | 0.40 ± 0.06 | 3 | p < 0.001 | *Figure 3B* |
| ATP | 37°C | – | 3.69 ± 0.49 | 7 | ns | *Figure 3C* |
| ATP+Vi | 37°C | – | 3.77 ± 0.38 | 4 | ns | *Figure 3C* |
| ATP+BeFx | 37°C | – | 3.20 ± 0.15 | 3 | ns | *Figure 3D* |
| ATP | 4°C | + | 4.38 ± 0.24 | 6 | ns | *Figure 3E* |
| ATP+Vi | 4°C | + | 4.01 ± 0.23 | 3 | ns | *Figure 3E* |
| ATP+BeFx | 4°C | + | 4.06 ± 0.46 | 3 | ns | *Figure 3F* |
| AMP-PNP | 37°C | + | 5.29 ± 0.22 | 2 | ns | *Figure 3G* |
| AMP-PNP+Vi | 37°C | + | 4.91 | 1 | nd | *Figure 3G* |
| AMP-PNP+BeFx | 37°C | + | 5.88 | 1 | nd | *Figure 3H* |
| ADP | 37°C | + | 7.37 ± 2.31 | 6 | p < 0.001 | *Figure 3I* |
| ADP+Vi | 37°C | + | 0.44 ± 0.10 | 3 | p < 0.001 | *Figure 3I* |
| ADP+BeFx | 37°C | + | 0.57 ± 0.32 | 3 | p < 0.001 | *Figure 3J* |

The online version of this article includes the following source data for table 1:

**Source data 1.** Source Data to *Table 1*.

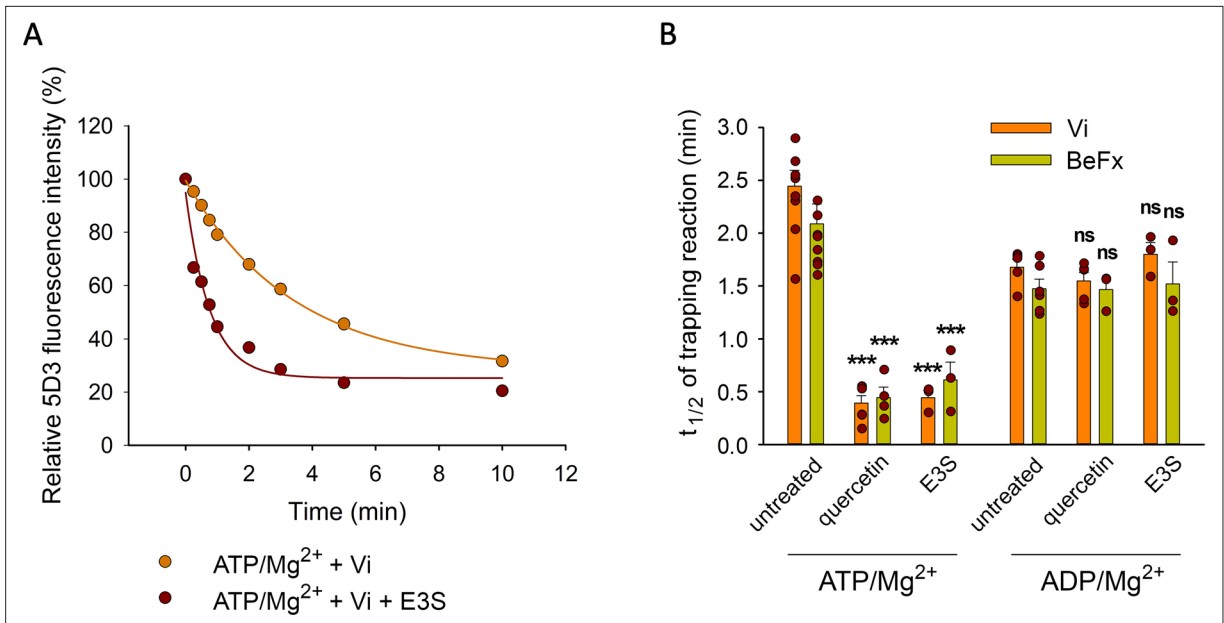

**Figure 4.** Effect of transported substrates on the kinetics of the formation of the Vi- and BeFx-trapped complexes. Permeabilized MDCK-ABCG2 cells were pre-treated or not with 10 µM quercetin or E3S for 10 min at 37°C and then further incubated with 0.5 mM ATP/Mg$^{2+}$ or ADP/Mg$^{2+}$ in the presence of Vi or BeFx. Samples taken at different time points were stained with 5 µg/ml 5D3-A647 on ice for 45 min. Panel (**A**) shows a representative Vi-trapping experiment in the absence or presence of E3S, while panel (**B**) summarizes the $t_{1/2}$ values calculated from the exponential fit of the kinetic curves (see Materials and methods). Mean ± SD of 3–5 independent experiments is shown. Significant differences compared to substrate untreated samples are shown by ***: $p < 0.001$.

The online version of this article includes the following source data for figure 4:

**Source data 1.** Source Data to *Figure 4B*.

(nucleotide-free) MDCK cells expressing ABCG2 were pre-labeled with 5D3-A647 antibody. Unbound 5D3-A647 molecules were removed, and cells were incubated at 37°C in a sufficiently large volume to prevent rebinding of the antibody. Under these conditions, we observed a gradual decrease of the 5D3-A647 fluorescence of cells, which was completely prevented by Ko143 treatment, supporting the notion that the ABCG2 molecules are intrinsically dynamic, while Ko143 stabilizes them in the IF 5D3-reactive conformation (*Figure 5A*). The dissociation rate of the antibody was significantly enhanced in the presence of transported substrates or ATP/Mg$^{2+}$ (*Figure 5A and B*). However, the largest (about fivefold) decrease of the $t_{1/2}$ values corresponding to the half-life of the 5D3-bound ABCG2 conformer was observed when substrates were co-administered with ATP/Mg$^{2+}$ (*Figure 5B*).

In further experiments, intracellular nucleotide pools were replenished with a non-hydrolyzable ATP analog AMP-PNP/Mg$^{2+}$ (*Figure 5B*). In the absence of ATP hydrolysis, ABCG2 molecules undergo IF to OF transition (*Figure 3G and H*), and the backward transition to the IF state has an extremely low probability. The time dependence of the AMP-PNP/Mg$^{2+}$-induced 5D3 dissociation was comparable to that obtained with ATP/Mg$^{2+}$ supporting that the antibody dissociation kinetics observed either in the presence of ATP/Mg$^{2+}$ or AMP-PNP/Mg$^{2+}$ may reflect the first nucleotide-induced IF to OF transition of ABCG2 (*Figure 5B*). Strikingly, when co-administered with AMP-PNP/Mg$^{2+}$, the same substrates did not increase further the dissociation rate of 5D3, suggesting that the NBD dimer formation induced by AMP-PNP/Mg$^{2+}$ binding switches ABCG2 into the low drug binding affinity state (*Figure 5B*). However, a similar decrease of the $t_{1/2}$ values was observed when substrates were added before AMP-PNP/Mg$^{2+}$ treatment (compare *Figure 5B and C*), indicating that transition to the OF state can also be accelerated by substrates with AMP-PNP/Mg$^{2+}$ treatment. Collectively, the above data imply that substrate binding at the TMDs induces a structural change in the transporter that can facilitate the nucleotide-dependent NBD dimer formation and the concomitant IF to OF transition, probably by reducing the energy barrier of the above conformational changes (*Orelle et al., 2022*).

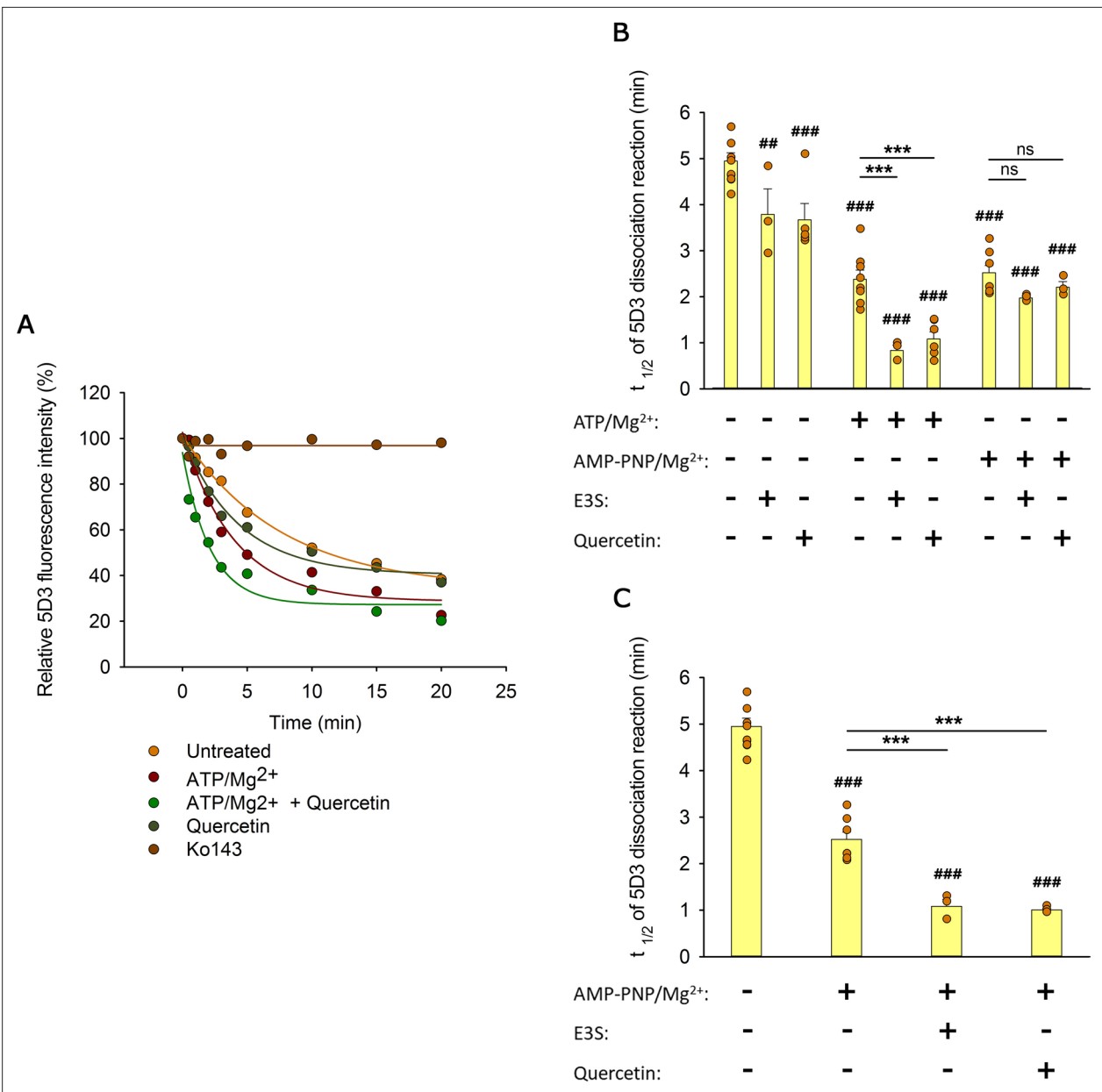

**Figure 5.** Effects of substrates and nucleotides on the kinetics of 5D3 dissociation when substrates were added simultaneously with nucleotides (**A, B**), or prior to nucleotides (**C**). Permeabilized MDCK-ABCG2 cells were pre-labeled with 5D3-A647 in the absence (panels (**A**) and (**B**)) or presence (panel (**C**)) of substrates (10 μM quercetin or 10 μM E3S) for 20 min at 37°C. After removal of the unbound antibody, cells were further incubated with 3 mM ATP/Mg$^{2+}$ or AMP-PNP/Mg$^{2+}$ in the absence or presence of substrates. Panel (**A**) shows representative 5D3 dissociation curves in the indicated conditions, while panels (**B**) and (**C**) summarizes the $t_{1/2}$ values calculated from the exponential fit of the dissociation curves (see Materials and methods). Means ± SD of 3–5 independent experiments are shown. Significant differences compared to untreated samples (first column) are shown by ###: $p < 0.001$ or ##: $p < 0.01$. Significant differences compared to only nucleotide-treated samples are indicated by ***: $p < 0.001$.

The online version of this article includes the following source data for figure 5:

**Source data 1.** Source Data to *Figure 5B and C*.

## The nucleotide-free IF conformation of ABCG2 has higher substrate affinity compared to the Vi-trapped post-hydrolytic conformation

In the following experiments, we visualized the subcellular localization of ABCG2-GFP and the fluorescent ABCG2 substrate mitoxantrone (MX) by using confocal microscopy. In accordance with previous observations (*Homolya et al., 2011*), at low concentrations, MX only stained poorly the MDCK

ABCG2-GFP cells, while ATP depletion, Ko143 or Vi treatments increased the intracellular accumulation of MX (*Figure 6A and B*). Interestingly, ATP-depleted cells exhibited strong plasma membrane staining by MX (*Figure 6A*). Plasma membrane staining by MX in both native and ATP-depleted cells was abolished by treatment with the competitive inhibitor Ko143, suggesting that red fluorescence in the plasma membrane of ATP-depleted cells reflects MX binding to ABCG2 molecules. To quantify the fraction of MX-bound ABCG2 molecules, we calculated the Pearson's correlation coefficients (PCC) between the MX and ABCG2-GFP signals in pixels representing the plasma membrane. Since the ABCG2-GFP signal was unchanged during the course of the different treatments (*Figure 6C*), the correlation coefficients depend mostly on MX binding to the transporter. In ATP-depleted cells, the high correlation values indicate that the majority of ABCG2 molecules reside in an MX-bound conformation (PCC = 0.72 ± 0.12) (*Figure 6D*). The correlation between the two signals strongly decreased in the presence of Ko143 (PCC = –0.1 ± 0.18), suggesting displacement of MX from the substrate binding site of the transporter by the competitive inhibitor. Similar results were obtained when ATP depletion was combined with Ko143 treatment (PCC = –0.04 ± 0.17). Binding of MX to ABCG2 was also suppressed by Vi (PCC = 0.18 ± 0.13), which is consistent with the notion that the post-hydrolytic ABCG2 conformer possesses low substrate affinity. Interestingly, in untreated cells, we measured significantly higher co-localization between the MX and ABCG2-GFP signals (PCC = 0.3 ± 0.12) than in Ko143-treated cells, suggesting that in the plasma membrane of live cells, a significant subset of ABCG2 molecules resides in an MX-bound IF state.

## MX binding to ABCG2 is confirmed by its reduced mobility using FCS measurements

As an independent approach to follow MX binding to ABCG2, we measured the mobility of MX in the plasma membrane by FCS. MX molecules bound to ABCG2 are expected to show decreased diffusion compared to free MX (*Horsey et al., 2020*). We analyzed the fluorescence autocorrelation functions (ACFs) of MX (orange) and ABCG2-GFP (green) in the plasma membrane using a two-component model (*Figure 7A*). Upon ATP depletion, the diffusion coefficient of MX decreased to the level obtained for ABCG2-GFP (*Figure 7B*), indicating that MX molecules readily bind to the nucleotide-free IF conformer of ABCG2. In accordance with the data obtained from cellular distributions (*Figure 6A and D*), the competitive inhibitor Ko143 prevented MX binding to ABCG2, resulting in the dominance of a high plasma membrane mobility MX population similar to the only MX-treated cells (*Figure 7B*).

## AMP-PNP binding switches ABCG2 to a conformation that is unable to bind MX

As shown in *Figure 3*, permeabilization of cells with streptolysin-O (SLO) synchronizes ABCG2 molecules in a nucleotide-free, 5D3-reactive IF conformation also observed in cryo-EM structures (*Taylor et al., 2017*). When permeabilized cells were treated with MX alone, we measured a strong co-localization between ABCG2-GFP and MX in the plasma membrane (PCC = 0.85 ± 0.05), confirming the high substrate affinity of the IF conformation of ABCG2. Strikingly, pre-incubation of permeabilized cells with 5 mM AMP-PNP/Mg$^{2+}$ strongly reduced the co-localization between MX and ABCG2-GFP in the plasma membrane (PCC = –0.1181 ± 0.2019), indicating that the conformational changes induced by AMP-PNP binding prevented MX binding to ABCG2 (*Figure 8*).

## Discussion

According to the alternating access model formulated by Jardetzky more than 50 years ago, membrane transporters alternate between IF and OF states, in which the centrally located substrate-binding site is accessible to only one side of the membrane at a time (*Jardetzky, 1966*). In active transporters, such as ABC transporters, accessibility changes are accompanied by a significant change of substrate affinity that is linked to binding and hydrolysis of ATP (*Seeger and van Veen, 2009*). Based on homology models and cryo-EM structures, ABCG2 is believed to alternate between a nucleotide-free IF and a nucleotide-bound OF conformation during its transport cycle (*Figure 1A and B*). The switch between the NBD-dissociated IF and the NBD-associated OF states involves a series of conformational changes that finally result in the uphill transport of substrates. In agreement with cryo-EM

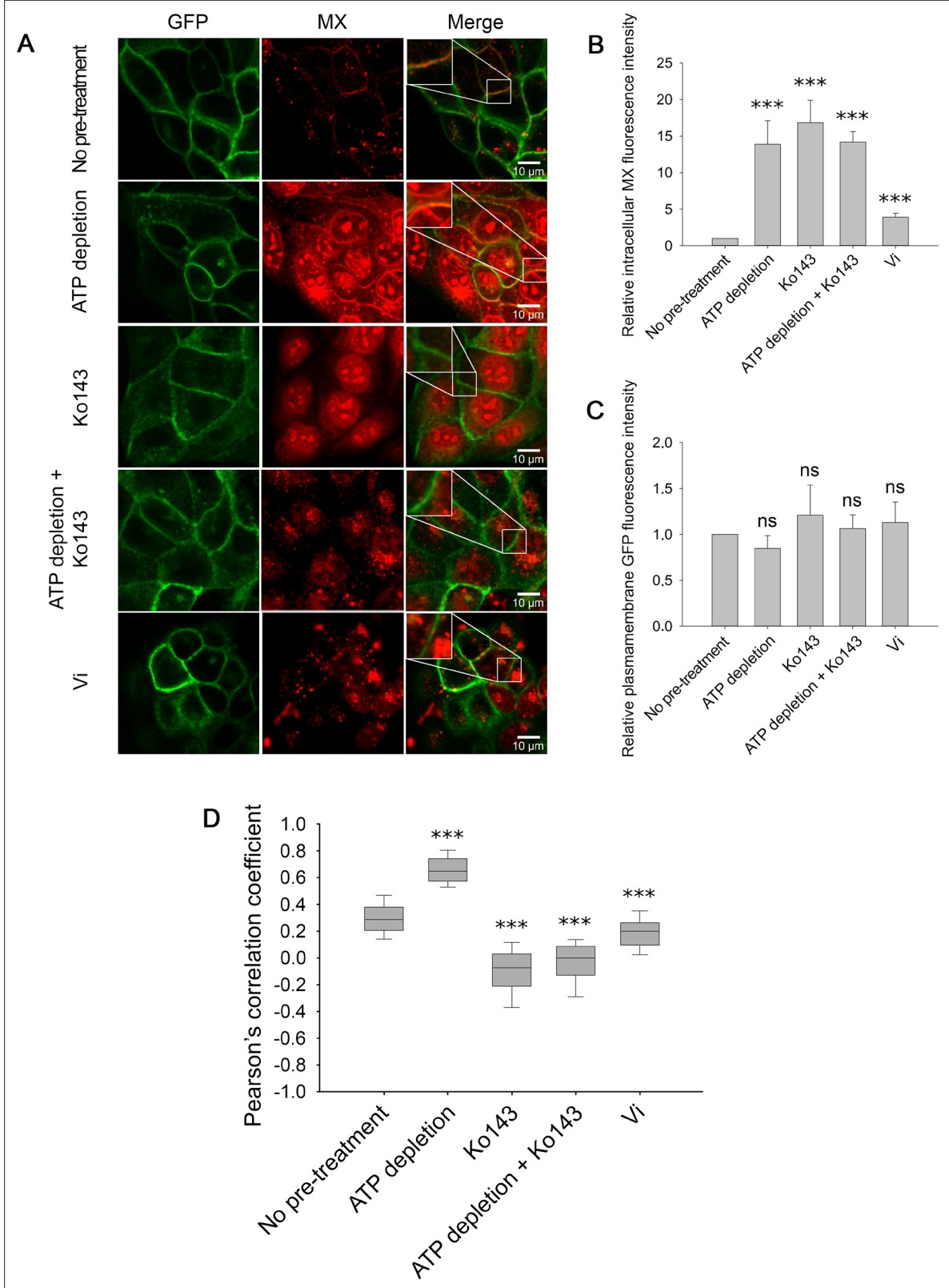

**Figure 6.** Cellular distribution of MX (red) in MDCK cells expressing ABCG2-GFP (green) in response to ATP depletion and/or Ko143 or Vi treatment (**A**). All treatments increased the intracellular MX fluorescence intensity (**B**), while the ABCG2-GFP fluorescence intensity in the plasma membrane remained unchanged (**C**). Pearson's correlation coefficients between the intensity distributions of MX and ABCG2-GFP in the plasma membrane pixels reveal the stabilization of a high-affinity substrate binding ABCG2 conformation in energy-deprived cells (**D**). ATP depletion was induced by 15 min pre-treatment

*Figure 6 continued on next page*

*Figure 6 continued*

of cells with 10 mM sodium azide and 8 mM 2-deoxy-D-glucose. Ko143, a non-fluorescent, competitive ABCG2 inhibitor, was added 10 min before MX staining. In panels (**B**) and (**C**), bars represent mean ± SD values, while panel (**D**) shows box and whisker plots. For each treatment group, 150–200 cells were analyzed from three to five independent experiments. Significant differences compared to the untreated control are shown by \*\*\*: p < 0.001.

The online version of this article includes the following source data for figure 6:

**Source data 1.** Source Data to *Figure 6B–D*.

studies, we show in nucleotide titration experiments (*Figure 3*) that 5D3 exclusively recognizes the nucleotide-free IF conformation of ABCG2 (*Taylor et al., 2017*), as the 5D3 recognized epitope of ABCG2 only forms in the IF conformation, while it is disrupted in the OF conformation (*Figure 1C–F*).

While semi-permeabilized cells provide a unique tool to study ABCG2 in a quasi-natural environment, the antibody-shift assay has several limitations. Because of the conformation selectivity of 5D3, details of the allosteric coupling between the NBDs and TMDs are inferred based on the influence of nucleotides and substrates on the population of ABCG2 molecules in the IF state, without any direct information on the intermediate or OF sates. Still, by modulating the levels of various nucleotide species, we were able to delineate distinct steps of the ATPase cycle, such as nucleotide binding or ATP hydrolysis. Moreover, we characterized MX binding to ABCG2 to better understand the cross-talk between nucleotide and substrate binding. While the interactions of fluorescent substrates with purified ABC transporters have been studied in nanodiscs (*Li et al., 2017*) or styrene maleic acid lipid copolymer particles (*Horsey et al., 2020*), to our knowledge we have applied the FCS technique for the first time to study substrate binding to ABCG2 in its natural plasma membrane environment in live cells.

In accordance with structural studies, our confocal microscopy experiments carried out in live cells show that the *apo* form of ABCG2 possesses high MX-affinity, while nucleotide binding and the concomitant dimerization of the NBDs induce conformation changes that prevent MX-binding. The

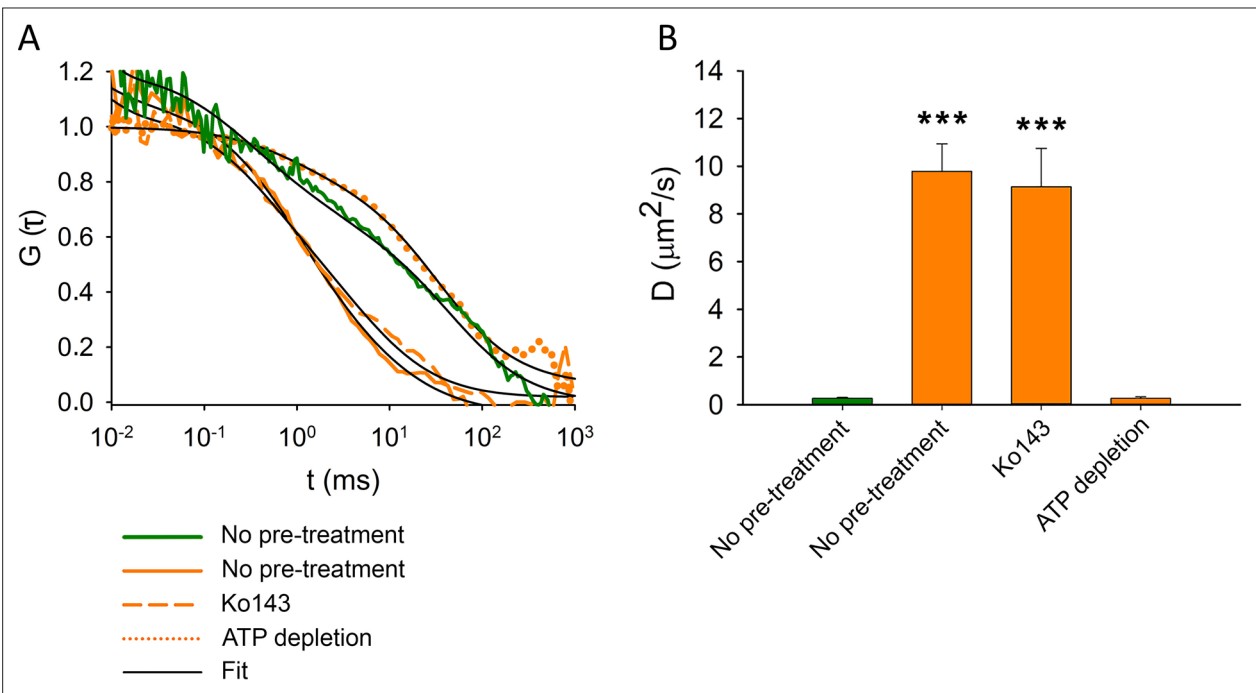

**Figure 7.** Fluorescence autocorrelation functions (**A**) and diffusion constants (**B**) of ABCG2-GFP (green) and MX (orange) in the plasma membrane of intact MDCK cells. ABCG2-GFP expressing cells without pre-treatments or following ATP depletion or Ko143 pre-treatment were stained with 100 nM MX for 15 min at 37°C. Each bar shows the mean ± SD for n = 50–100 cells from at least three independent measurements. Significant differences compared to the diffusion constant of ABCG2-GFP (green bar) in the plasma membrane of untreated cells are shown by \*\*\*: p < 0.001.

The online version of this article includes the following source data for figure 7:

**Source data 1.** Source Data to *Figure 7B*.

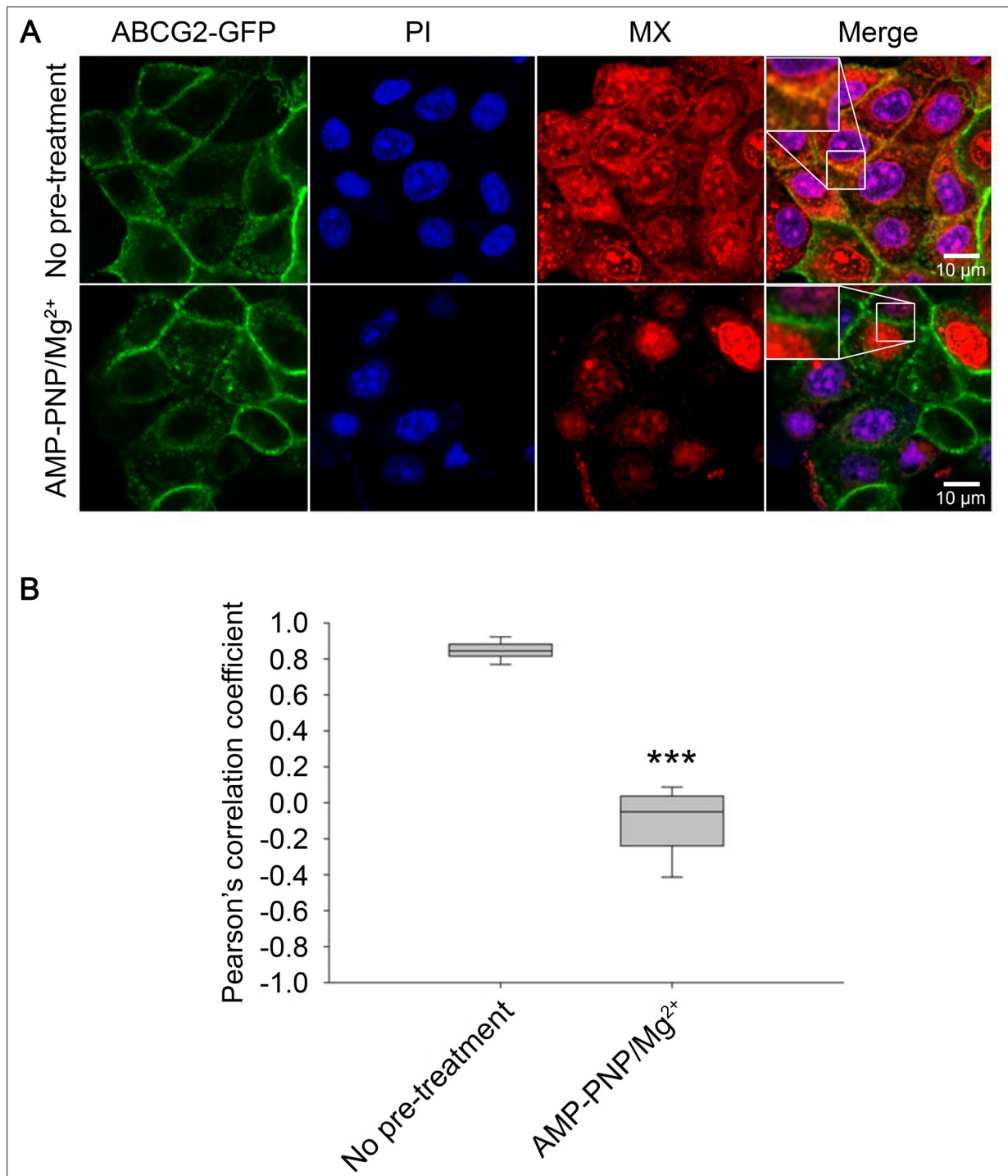

**Figure 8.** Substrate affinity changes of ABCG2 in permeabilized cells. SLO-permeabilized ABCG2-GFP expressing cells were pre-treated or not with 5 mM AMP-PNP/Mg$^{2+}$ for 15 min and then stained with 500 nM MX for 15 min at 37°C. Permeabilized cells were identified by PI staining. Box and whisker plots show Pearson's correlation coefficients of n > 150 cells from three independent experiments. ***p < 0.001 by Kolmogorov-Smirnov test. MX, mitoxantrone; PI, propidium iodide; SLO, streptolysin-O.

The online version of this article includes the following source data for figure 8:

**Source data 1.** Source Data to *Figure 8B*.

simultaneous drop in the 5D3- and MX-binding indicates that the high-to-low switch in drug binding affinity coincides with the transition from the IF to the OF conformation. Ko143, which can displace MX from the substrate binding site (see *Figures 6 and 7*) also prevents the transition of ABCG2 from the IF to the OF conformation (*Figure 5A*) in agreement with previous cryo-EM studies (*Orlando and Liao, 2020*; *Yu et al., 2021*; *Jackson et al., 2018*). Our data also directly indicate that high-affinity binding of MX requires the IF conformation and that energizing the transporter by nucleotides is not required for substrate binding. The nucleotide titration experiments prove that nucleotide binding (ATP, AMP-PNP, and ADP) and concomitant NBD dimer formation is sufficient to induce the conformational switch to the OF conformation, and that ATP hydrolysis is not essential for the nucleotide-dependent IF-OF transition (*Figure 3*) and the high-to-low switch in drug binding affinity (*Figure 8*), supporting the results of structural studies carried out using the catalytic glutamate mutant ABCG2 variant (*Manolaridis et al., 2018*). We also observed that in the presence of physiological ATP concentrations, in the context of the plasma membrane of live cells, a small, but significant fraction of ABCG2 molecules reside in the MX-bound state (*Figure 6D*) ready to initiate a productive transport cycle. The $V_i$-trapped post-hydrolysis state possesses low substrate affinity (*Figure 6A and D*), confirming that dissociation of the hydrolysis products is required to switch the transporter from the OF conformation back to the IF conformation and to reset its high substrate affinity.

Previous studies observed analogous changes of the substrate binding affinity in ABCB1 (*Martin et al., 2001*), ABCC1 (*Rothnie et al., 2006*), and ABCG2 (*McDevitt et al., 2008*) using radioligand binding assays. ABCB1 exists in an equilibrium between OF and IF states, which are readily interconverted by ligand binding (*Marcoux et al., 2013*). A similar, bidirectional interdomain crosstalk between the NBDs and TMDs was observed in human ABCB1 using the UIC2-reactivity assay (*Bársony et al., 2016*; *Goda et al., 2020*). The fact that the antibody binding site and the NBDs of the transporters are ~80 Å apart and on opposite sides of the membrane suggests that long-range conformational couplings between the TMD and NBD motions are conserved features among human ABC exporters. However, the similarity in coupling of the ATPase activity and the transport cycles of ABCB1 and ABCG2 is remarkable in view of the differences in the conformational changes of the TMDs and the substrate binding cavities (*Arana and Altenberg, 2019*; *van Wonderen et al., 2014*; *Locher, 2016*), and the different modes of the nucleotide-dependent NBD motions (*Stockner et al., 2020*).

Like several other ABC transporters, ABCG2 possesses significant basal ATPase activity, which is generally increased 2- to 5-fold by transported substrates (*Telbisz et al., 2012*). The increased catalytic turnover is reflected by the enhanced steady-state ATP hydrolysis rate (*Telbisz et al., 2012*; *Ozvegy et al., 2001*) and the increased rate at which ABCG2 becomes trapped in the Vi- or BeFx-bound post-hydrolytic states (*Figure 4*).

Our experiments demonstrated that 5D3 binding to ABCG2 is reversible, and addition of substrates, ATP/$Mg^{2+}$ or their combination can accelerate the kinetics of antibody dissociation. The reversible nature of 5D3 binding also explains previous observations that treatments with saturating concentration of the antibody did not induce complete inhibition of transport and of ATPase activity (*Ozvegy-Laczka et al., 2005*).

Importantly, 5D3 dissociation experiments carried out in the absence of nucleotides clearly showed that substrate binding alone has an effect on the conformation of ABCG2 (*Figure 5A and B*). When substrates are presented prior to the nucleotides, the IF to OF transition is accelerated. In contrast, when the nucleotide analog AMP-PNP is allowed to bind first, substrates cannot promote the transition (*Figure 5B and C*). Thus, substrates accelerate the cumulative step of nucleotide binding, NBD 'sandwich-dimer' formation, and the concomitant IF to OF transition, but only when having access to the substrate binding site as present in the IF state. The substrate-induced conformational change might be small as indicated by a recent study, which could not detect these changes as an increase in the proximity ratio of FRET between the NBDs of ABCB1 in permeabilized cells deprived of ATP/$Mg^{2+}$ (*Futamata et al., 2020*). Accordingly, in recent cryo-EM studies carried out at turnover conditions (in the presence of substrate and ATP/$Mg^{2+}$), two conformers representing the transition from the IF state to a semi-closed state were identified. These structures revealed that the accessibility of the substrate binding site gradually decreases upon the closure of the NBD dimer (*Yu et al., 2021*), and therefore, substrates should bind to the IF conformer.

In conclusion, our results indicate that nucleotide binding is the major regulator of TMD conformation in ABCG2. The nucleotide induced IF to OF transition coincides with the high-to-low switch

of substrate affinity, and this event precedes ATP hydrolysis. Detailed kinetic analysis of several ABC transporters will be needed to establish differences and similarities of the catalytic mechanisms within the superfamily. In the case of the channel-type ABC protein CFTR (ABCC7), pore opening, believed to correspond to the IF to OF transition, represents the slowest step of the gating cycle (*Vergani et al., 2013*). In contrast, in the case of ABCC1 (*Wang et al., 2020*) and ABCB1 (*Bársony et al., 2016*), the IF to OF transition is not the rate-limiting step of the catalytic cycle. Similarly, *Figure 3A and B* reveals that in the presence of saturating concentrations of hydrolyzable ATP, the majority of ABCG2 molecules adopt the 5D3-dim OF conformation, indicating that the IF to OF transition is not the slowest step of the catalytic cycle and consequently it is not the major determinant of the cycle time. Accordingly, in live cells, only a small fraction of ABCG2 molecules is in the substrate-bound IF conformation ready to harvest the energy of ATP for transport, while depletion of ATP increases the proportion of ABCG2 molecules in the IF conformation by prolonging the waiting time before the IF to OF transition (*Figure 6A and D*). How can we explain the stimulation of the steady-state ABCG2 ATPase activity by transported substrates? On the one hand, results shown in *Figure 5* clearly demonstrate that bound substrates accelerate the IF to OF transition, probably by facilitating the nucleotide-dependent NBD dimer formation. However, a mere speeding of the IF to OF transition cannot significantly increase the turnover rate, suggesting that substrate-mediated stimulation of the ATPase must reflect acceleration of some other step(s). At the same time, results presented in *Figures 6–8* indicate that substrates can no longer bind to the OF, low-substrate-affinity conformation, and therefore their influence on the rate constants of ATP hydrolysis and the concomitant NBD dissociation can be ruled out. This apparent paradox may be resolved by assuming two distinct pre-hydrolytic OF states associated with an uncoupled, basal activity and a coupled, drug transport dependent ATPase activity (*Al-Shawi et al., 2003*; *Doshi and van Veen, 2013*). In the coupled activity pathway, transported substrates increase the catalytic rate of ABCG2 by accelerating the IF-OF transition similar to ABCC1 (*Wang et al., 2020*). In the absence of drugs, the transporter follows a different catalytic path, in which the step limiting the uncoupled ATPase activity follows the IF-OF transition. Eventually, ABCG2 molecules that have reached the NBD 'sandwich-dimer' without substrate binding either dissociate without hydrolysis, or undergo a futile ATP hydrolysis cycle to reset the transporter into an IF state with high drug binding affinity. In the future, single molecule-based approaches (*Catipovic et al., 2019*; *Josts et al., 2020*; *Wang et al., 2020*) may provide further insights into the kinetics of the above transitions.

## Materials and methods

### Key resources table

| Reagent type or resource | Designation | Source or reference | Identifiers | Additional information |
|---|---|---|---|---|
| Antibody | 5D3 (mouse monoclonal) | Hybridoma was donated by Brian P. Sorrentino | | Prepared from hybridoma supernatant by affinity chromatography |
| Antibody | BXP-21 (mouse monoclonal) | Santa Cruz Biotechnology | Cat# sc-58222 | (1:2500 dilution) |
| Antibody | C-2 (mouse monoclonal) | Santa Cruz Biotechnology | Cat# sc-8432 | (1:2500 dilution) |
| Antibody | Goat anti-mouse IgG-HRP (polyclonal) | Santa Cruz Biotechnology | Cat# sc-2005 | (1:2500 dilution) |
| Cell line (*Canis familiaris*, dog) | MDCK II, (epithelial-like cells from kidney distal tubule) | Obtained from Prof. Gerrit van Meer | ECACC 00062107 | Mycoplasma-free |
| Cell line (*Mus musculus*, mouse) | 5D3 hybridoma | Donated by Brian P. Sorrentino | | Mycoplasma-free |
| Chemical compound, drug | Mitoxantrone | Sigma-Aldrich | Cat# M6545 | |
| Chemical compound, drug | Ko143 | Sigma-Aldrich | Cat# K2145 | |
| Chemical compound, drug | Quercetin | Sigma-Aldrich | Cat# Q4951 | |

*Continued on next page*

*Continued*

| Reagent type or resource | Designation | Source or reference | Identifiers | Additional information |
|---|---|---|---|---|
| Chemical compound, drug | Estrone-3-sulfate | Sigma-Aldrich | Cat# E9145 | |
| Chemical compound, drug | ADP | Sigma-Aldrich | Cat# A2754 | |
| Chemical compound, drug | AMP-PNP | Sigma-Aldrich | Cat# A2647 | |
| Chemical compound, drug | ATP | Sigma-Aldrich | Cat# A2383 | |
| Chemical compound, drug | Sodium-orthovanadate | Sigma-Aldrich | Cat# S6508 | |
| Chemical compound, drug | Protease Inhibitor Cocktail | Sigma-Aldrich | Cat# P2714 | |
| Chemical compound, drug | DL-Dithiothreitol | Sigma-Aldrich | Cat# D0632 | |
| Software, algorithm | Flowing software | Turku Centre for Biotechnology | | https://bioscience.fi/services/cell-imaging/flowing-software/ |
| Software, algorithm | MATLAB | Mathworks Inc. | | https://www.mathworks.com/products/matlab.html |
| Software, algorithm | QuickFit 3.0 | | | https://biii.eu/quickfit-3 |
| Software, algorithm | SigmaPlot | Systat Software Inc. | | https://systatsoftware.com/sigmaplot/ |
| Other | Eight-well chambered coverslip plate | ibidi GmbH | Cat# 80826-90 | Imaging chamber for microscopy |
| Other | Streptolysin-O (SLO) from Streptococcus pyogenes | Sigma-Aldrich | Cat# S5265 | (250 U/ml) |
| Other | Alexa 647 succinimidyl ester | Life Technologies Inc. | Cat# A20006 | Fluorescent dye |
| Other | Propidium iodide | Sigma-Aldrich | Cat# P4170 | Fluorescent dye |
| Other | SuperSignal West Pico PLUS Chemiluminescent Substrate | Thermo Fisher Scientific | Cat# 34579 | Reagent for protein detection in western blot analysis |

## Chemicals

Cell culture media, supplements, and chemicals were purchased from Sigma-Aldrich (Budapest, Hungary). Alexa 647 succinimidyl ester (A647) was purchased from Life Technologies, Inc (Carlsbad, CA). The 5D3 anti-ABCG2 mAb was prepared from hybridoma supernatants by affinity chromatography. Purity (>97%) was verified by SDS/PAGE. The 5D3 hybridoma cell line was a kind gift from Brian P. Sorrentino (Division of Experimental Hematology, Department of Hematology/Oncology, St. Jude Children's Research Hospital, Memphis, TN). 5D3 antibody was labeled with A647 (5D3-A647) and was separated from the unconjugated dye by gel filtration using a Sephadex G-50 column. The dye-to-protein ratio was approximately 3 for each antibody preparation. Stock solutions of nucleotides were prepared in distilled water at pH 7 (by Tris-Base).

## Cell lines

The MDCK II Madin-Darby canine kidney cell line was a kind gift from Gerrit van Meer (Department of Membrane Enzymology, Centre for Biomembranes and Lipid Enzymology Utrecht, The Netherlands). The MDCK II cell lines stably expressing ABCG2 or its N-terminally green fluorescent protein (GFP) tagged variant (*Orbán et al., 2008*) were established using the *Sleeping Beauty* transposon-based gene delivery system (*Erdei et al., 2018*). Cells expressing the transgene at high level were selected based on their 5D3-A647 or GFP fluorescence by repeated flow cytometry sorting using a Becton Dickinson FACSAria III Cell Sorter (Becton Dickinson, Mountain View, CA). Cells were grown as monolayer cultures in Dulbecco's modified Eagle's medium supplemented with 0.1 mg/ml penicillin-streptomycin

cocktail, 10% heat-inactivated fetal calf serum and 2 mM L-glutamine. Cells were maintained at 37°C in a 5% $CO_2$ atmosphere and were grown to approximately 80% confluency. Cells were regularly checked for Mycoplasma infection and were found to be negative.

## Western blot analysis

Cells ($2 \times 10^5$) were lysed in 100 µl reducing Laemmli sample buffer (6×) for 10 min at 95°C. Afterward, the lysates were subjected to SDS-polyacrylamide gel electrophoresis using an 8% polyacrylamide gel and then electroblotted onto a nitrocellulose membrane with a pore size of 0.45 µm (GE Healthcare Life Sciences, Little Chalfont, Buckinghamshire, UK). ABCG2 expression was detected by the BXP-21 mouse mAb, while actin was labeled with the C-2 mouse mAb (both from Santa Cruz Biotechnology Inc, Santa Cruz Biotechnology, CA). As a secondary antibody, a goat anti-mouse HRP-conjugated IgG (Santa Cruz Biotechnology Inc, Santa Cruz, CA) was applied. All antibodies were used at 1:2500 dilution. Bands were visualized with SuperSignal West Pico PLUS Chemiluminescent Substrate (Thermo Fisher Scientific, Waltham, MA) using the FluorChem Q gel documentation system (Alpha Innotech Corp, San Leandro, CA).

## MX accumulation test

The transport activity of ABCG2 and ABCG2-GFP was studied using an MX accumulation assay (*Tarapcsák et al., 2017*). Cells ($0.5 \times 10^6$ ml$^{-1}$ in PBS containing 7 mM glucose [gl-PBS]) were pre-incubated in the presence or absence of 2 µM Ko143 for 15 min at 37°C and then stained with 5 µM MX for 30 min. Samples were washed three times with ice-cold gl-PBS containing 0.5% fetal bovine serum (FBS) and stored on ice until flow cytometry measurement. To exclude dead cells from the analysis, samples were stained with propidium iodide (PI).

## 5D3-reactivity assay

Cells ($0.5 \times 10^6$ ml$^{-1}$ in gl-PBS) were pre-incubated with or without 2 µM Ko143 for 10 min and then further incubated with 5 µg/ml 5D3-A647 monoclonal anti-ABCG2 antibody at 37°C. After 30 min of incubation, samples were washed two times with ice-cold gl-PBS and centrifuged for 5 min at 435×*g* at 4°C. The 5D3-A647 fluorescence intensity of the cells was measured by flow cytometry.

## Permeabilization of cells with SLO toxin

SLO (Sigma-Aldrich, Budapest, Hungary) is a pore-forming exotoxin from *Streptococcus pyogenes*. The SLO pores formed in the membrane are permeable to small water-soluble molecules including nucleotides (*Yang et al., 2006*). SLO is an oxygen-labile toxin that is reversibly activated by dithiothreitol (DTT). Cell suspensions ($1 \times 10^7$ cells/ml) were treated with 250 U/ml SLO in the presence of 1 mM DTT, Protease Inhibitor Cocktail (2 mM AEBSF, 0.3 µM aprotinin, 116 µM bestatin, 14 µM E-64, and 1 µM leupeptin), 0.5 mM PMSF, and 1% FBS in gl-PBS at 37°C for 30 min, which allowed permeabilization of approximately 50% of cells, as it was verified by PI staining. The reaction was stopped with 20 ml PBS containing 1% FBS and the cells were centrifuged at 635×*g* for 5 min at room temperature. Unbound toxin was removed by washing the cells three times with PBS and the cell pellet was resuspended in PBS (*Goda et al., 2020*). The applied 1 mM DTT concentration did not affect the 5D3-reactivity of ABCG2.

Cells grown in eight-well chambered coverslip plates (ibidi GmbH, Gräfelfing, Germany) for confocal microscopy experiments were permeabilized using 62.5 U/ml SLO in the presence of 1 mM DTT and Protease Inhibitor Cocktail at 37°C for 15 min, in HEPES solution (20 mM HEPES, 123 mM NaCl, 5 mM KCl, 1.5 mM $MgCl_2$, and 1 mM $CaCl_2$) containing 1% FBS.

## Determination of apparent affinity of nucleotide binding

Apparent affinity of nucleotide binding ($K_A$) was determined as described previously (*Bársony et al., 2016*; *Goda et al., 2020*). Permeabilized cells ($1 \times 10^6$ ml$^{-1}$) were pre-treated with nucleotides added at different concentrations in the presence of equimolar concentrations of $Mg^{2+}$ at 37°C for 10 min and then further incubated with 5 µg/ml 5D3-A647 at 37°C for 20 min. To prevent ATP hydrolysis, ATP was added without $Mg^{2+}$ in the presence of 5 mM EDTA or the whole experiment was carried out on ice. In nucleotide trapping experiments, nucleotide treatments were applied together with 0.5 mM sodium orthovanadate (Vi) or BeFx (200 µM $BeSO_4$ and 1 mM NaF) at 37°C for 30 min. Subsequently, the cells

were labeled with 5 µg/ml 5D3-A647 on ice for 45 min after removal of the un-trapped nucleotides by washing them two times with ice-cold PBS. After antibody labeling samples were washed again three times with ice-cold PBS and centrifuged for 5 min at 635×g at 4°C. The mean 5D3-A647 fluorescence intensity of the cells was determined by flow cytometry and plotted as a function of the nucleotide concentration. To determine the apparent affinity of ABCG2 for nucleotides ($K_A$), data points were fitted with the four-parameter Hill function, where the $F_{min}$ and $F_{max}$ values represent the minimum and maximum fluorescence intensities, respectively:

$$F = \frac{F_{min} \times K_A^n + F_{max} \times x^n}{K_A^n + x^n} \qquad (1)$$

## Studying the kinetics of nucleotide trapping by Vi or BeFx

Permeabilized cells ($1 \times 10^6$ ml$^{-1}$) were incubated with 0.5 mM ATP/Mg$^{2+}$ or ADP/Mg$^{2+}$ and 0.5 mM $V_i$ or BeF$_x$ (200 µM BeSO$_4$ and 1 mM NaF) in the presence or absence of ABCG2 substrates (10 µM quercetin or 10 µM E3S) in PBS at 37°C. To follow the kinetics of the trapping reaction, 500 µl aliquots was taken at different time points and washed two times with 5 ml ice-cold PBS. After washing, the samples were resuspended in 500 µl ice-cold PBS and labeled with 5 µg/ml 5D3-A647 at 4°C for 45 min. The 5D3-A647 fluorescence intensity of the samples ($F$) was plotted as a function of time ($t$). The $t_{1/2}$ values, representing the half-life of the 5D3-reactive ABCG2 conformation, were calculated from an exponential fit of the data points according to the following equation:

$$F = F_0 \times e^{-t \times \frac{ln2}{t_{\frac{1}{2}}}} + c \qquad (2)$$

Wherein $F_0$ is the difference between the zero and infinite time points of the curve and $c$ is the background fluorescence intensity of cells.

## 5D3 dissociation

Permeabilized MDCK-ABCG2 cells ($1 \times 10^6$ ml$^{-1}$) were pre-labeled with 5D3-A647 in the presence or absence of 10 µM quercetin or 10 µM E3S for 20 min at 37°C. After removing the unbound 5D3-A647, cells ($1 \times 10^5$ ml$^{-1}$) were further incubated with 3 mM ATP/Mg$^{2+}$ or AMP-PNP/Mg$^{2+}$ in the absence or presence of the above substrates at 37°C. To study the kinetics of 5D3 dissociation, 500 µl aliquots were taken at regular intervals and washed two times with ice-cold PBS. The 5D3-A647 fluorescence intensity of the cells was measured by flow cytometry and plotted as a function of time (t). The $t_{1/2}$ values, representing the half-life of the 5D3-reactive ABCG2 conformation, were calculated from an exponential fit of the data points using *Equation 2*.

## Flow cytometry

Flow cytometry analysis was carried out using a Becton Dickinson FACS Array flow cytometer (Becton Dickinson, Mountain View, CA). A 635 nm laser was used for the excitation of MX and A647 and their fluorescence was detected in the red channel through a 661/16 nm bandpass filter, while a 532 nm laser was used for the excitation of PI and the emitted light was detected using a 585/42 nm band-pass filter. Cell debris was excluded from analysis on the basis of FSC and SSC signals. Fluorescence signals of $2 \times 10^5$ cells/sample were collected in logarithmic mode, and the cytofluorimetric data were analyzed using the Flowing software (Cell Imaging Core, Turku Centre for Biotechnology, Turku, Finland).

## Confocal laser scanning microscopy and fluorescence co-localization analysis

To assess the co-localization of ABCG2-GFP and the fluorescent ABCG2 substrate MX in the plasma membrane of MDCK cells, we carried out confocal laser scanning microscopy (CLSM) experiments. Measurements were performed in eight-well chambered coverslip plates (ibidi GmbH, Gräfelfing, Germany). ATP depletion of intact cells was induced by a 15 min pre-treatment with 8 mM 2-deoxy-D-glucose and 10 mM sodium azide in glucose-free medium. ATP-depleted or non-ATP-depleted cells were pre-treated with 2 µM Ko143 or 0.5 mM Vi for 15 min, stained with 500 nM MX for 15 min at 37°C and then washed three times with HEPES solution. SLO-permeabilized cells were

pre-stained with 6 µg/ml PI, then further incubated with 500 nM MX for 15 min at 37°C in the presence or absence of 5 mM AMP-PNP and subsequently washed three times with HEPES solution.

Fluorescence images were acquired with a Nikon A1 Eclipse Ti2 Confocal Laser-Scanning Microscope (Nikon, Tokyo, Japan) using a Plan Apo 60× water objective (NA = 1.27). Laser lines of 488 and 647 nm were used for the excitation of ABCG2-GFP and MX, while fluorescence emissions were detected through band pass filters of 500–550 and 660–740 nm, respectively. All the images were recorded with the same settings of the equipment, such as same high voltages, laser powers and pinhole. Images were acquired in sequential mode to minimize the crosstalk between channels. Images of approximately 1-µm-thick optical sections, each with 512 × 512 pixels, and a pixel size of approximately 200 nm, were acquired. A spatial averaging filter with a 3 × 3 mask was used to denoise the images. Co-localization analysis was carried out by calculating the PCCs between the pixel intensities of the two detection channels in pixels representing the plasma membrane (*Vámosi et al., 2004*). Only pixels where at least one of the intensities was above the threshold (2× the average autofluorescence intensity) were included in the analysis. Image analysis methods and routines were implemented in MATLAB scripts (Mathworks Inc, Natick, MA) (*Volkó et al., 2019*).

## Fluorescence correlation spectroscopy

To distinguish-free and ABCG2-bound MX molecules based on their different diffusion properties, FCS measurements were performed. FCS measurements were carried out using a Nikon A1 Eclipse Ti2 Confocal Laser-Scanning Microscope (Nikon, Tokyo, Japan), equipped with a Plan Apo 60×water objective (NA = 1.27) and a PicoQuant time-correlated single photon counting FCS (TCSPC-FCS) upgrade kit (PicoQuant, Berlin, Germany).

FCS measurements were carried out on live MDCK cells expressing ABCG2-GFP in eight-well chambered coverslip plates (ibidi GmbH, Gräfelfing, Germany). Cells were stained with 100 nM MX for 15 min at 37°C in the presence or absence of 2 µM Ko143 or after ATP depletion. Fluorescence of ABCG2-GFP and MX was excited with a 488 and a 647 nm laser, respectively. The fluorescence signals emitted by ABCG2-GFP and MX were detected in the spectral ranges of 500–550 and 660–740 nm using single photon counting detectors (PicoQuant, Berlin, Germany). Measurements of 10 × 10 s runs were taken at three selected points in the cross-section of the plasma membrane of each selected cell. Fluorescence autocorrelation curves were calculated using SymPhoTime64 software (PicoQuant, Berlin, Germany) at 200 time points from 300 ns to 1 s with a quasi-logarithmic time scale.

Autocorrelation curves of the doubly labeled cells were fitted to a triplet state model with two diffusion components to describe the 3D-diffusion of free MX (fast component) and the 2D diffusion of ABCG2-bound MX in the x-z plane of the plasma membrane (slow component). The laser beam was positioned in a region of the cell membrane parallel to the long axis of the ellipsoidal laser volume.

$$G\left(\tau\right) = \frac{1-T+Te^{\frac{-\tau}{\tau_{trip}}}}{N(1-T)}\left(\rho\frac{1}{1+\frac{\tau}{\tau_{D1}}}\frac{1}{\sqrt{1+\frac{\tau}{S^2\tau_{D1}}}} + \left(1-\rho\right)\frac{1}{\sqrt{1+\frac{\tau}{\tau_{D2}}}}\frac{1}{\sqrt{1+\frac{\tau}{S^2\tau_{D2}}}}\right) \qquad (3)$$

In *Equation 3*, N is the average number of fluorescent molecules in the detection volume, T is the fraction of molecules in the triplet state, $\tau_{trip}$ is the triplet correlation time. The diffusion rate is characterized by the diffusion time $\tau_D$, which is the average time spent by a molecule in the illuminated volume. $\tau_{D1}$ and $\tau_{D2}$ are the diffusion times of the fast and slow components, $\rho$ is the fraction of the first component, and $1-\rho$ is the fraction of the second component. The diffusion coefficients (*D*) of the fast and slow components were determined from the following equation:

$$D = \frac{\omega_{xy}^2}{4\tau_d} \qquad (4)$$

Wherein, $\omega_{xy}$ is the lateral $e^{-2}$ radius of the detection volume. $\omega_{xy}$ was measured by determining the diffusion time of 100 nM A647 dye (dissolved in 10 mM Tris, 0.1 mM EDTA-containing buffer, and pH 7.4) with known diffusion coefficient ($D_{A647}$ = 330 µm²/s, at T = 22.5°C) (*Weidemann and Schwille, 2013*) and substituting it into *Equation 4* that corresponds to the aspect ratio of the ellipsoidal confocal volume, defined as the ratio of its axial and radial dimensions. This parameter was estimated by fitting the autocorrelation curves of a 100 nM A647 dye solution.

## Statistical analysis and curve fitting

For the statistical analysis of data, SigmaPlot (version 14, SSI, San Jose, CA) was used. For the comparison of two samples from normally distributed populations with equal variances, Student's t-test was performed, while in case of unequal variances a Kolmogorov-Smirnov test was applied. Multiple comparisons were performed with analysis of variance applying the Holm-Sidak test for post hoc pair-wise comparison of the data. In the case of unequal variances, the Dunnett T3 post hoc pair-wise comparison method was used. Differences were considered significant at $p < 0.05$.

All curve fitting was carried out by SigmaPlot (version 14, SSI, San Jose, CA) except for fitting of autocorrelation curves that was performed by using the QuickFit 3.0 software developed in the group B040 (Prof. Jörg Langowski) at the German Cancer Research Center (DKFZ).

## Acknowledgements

The authors thank László Csanády for the critical reading of the manuscript and for helpful discussions, and Adél Vezendiné Nagy for her skilful technical assistance. We are grateful for the financial support by the Hungarian National Research, Development and Innovations Office (NKFIH; https://nkfih.gov.hu/english), grant number K124815 to KG, GINOP-2.2.1-15-2017-00079 to JR and KG, and HunProtEx 2018–1.2.1-NKP-2018–00005 and RRF-2.3.1-21-2022-00015 to GS. GS and TS was also supported by the Austrian Science Fund SFB35.

## Additional information

### Funding

| Funder | Grant reference number | Author |
| --- | --- | --- |
| National Research, Development and Innovation Office | K124815 | Katalin Goda |
| National Research, Development and Innovation Office | HunProtEx 2018-1.2.1-NKP-2018-00005 | Gergely Szakacs |
| National Research, Development and Innovation Office | RRF-2.3.1-21-2022-00015 | Gergely Szakacs |
| Austrian Science Fund | SFB35 | Gergely Szakacs Thomas Stockner |
| European Regional Development Fund | GINOP-2.2.1-15-2017-00079 | Judit Remenyik Katalin Goda |

The funders had no role in study design, data collection and interpretation, or the decision to submit the work for publication.

### Author contributions

Zsuzsanna Gyöngy, Formal analysis, Investigation, Visualization, Writing – original draft; Gábor Mocsár, Formal analysis, Investigation, Methodology; Éva Hegedűs, Zsuzsanna Ritter, Formal analysis, Investigation; Thomas Stockner, Conceptualization, Visualization, Writing – original draft, Writing – review and editing; László Homolya, Resources, Methodology, Writing – review and editing; Anita Schamberger, Investigation; Tamás I Orbán, Methodology; Judit Remenyik, Funding acquisition, Methodology; Gergely Szakacs, Conceptualization, Funding acquisition, Writing – original draft, Writing – review and editing; Katalin Goda, Conceptualization, Supervision, Funding acquisition, Methodology, Writing – original draft, Project administration, Writing – review and editing

### Author ORCIDs

Thomas Stockner http://orcid.org/0000-0002-7071-8283
Gergely Szakacs http://orcid.org/0000-0002-9311-7827
Katalin Goda http://orcid.org/0000-0003-2001-7400

Decision letter and Author response

Decision letter https://doi.org/10.7554/eLife.83976.sa1
Author response https://doi.org/10.7554/eLife.83976.sa2

---

## Additional files

### Supplementary files
• MDAR checklist

### Data availability
All data generated during this study are included in the manuscript. Source Data files have been provided for Table 1, Figure 2 and Figures 4 to 8.

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
