## [Editor Report]

The ABC transporter ABCG2 extrudes chemotherapy reagents and other xenobiotics from a number of different tissues. How ABCG2 operates at the molecular level has been largely derived from structures and dynamics carried out in non-physiological environments. The paper presents convincing cell-based evidence describing the relationship between structural changes of ABCG2 and substrate binding using flow cytometry, confocal microscopy, and fluorescence-correlation spectroscopy methods. Both the mechanistic conclusions and methodology employed offer important insights, which will be of general interest to the biochemistry and transport biology communities.

---

## [Decision Letter]

**Decision letter after peer review:**

Thank you for submitting your article "Nucleotide binding is the critical regulator of ABCG2 conformational transitions" for consideration by *eLife*. Your article has been reviewed by 3 peer reviewers, including David Drew as the Reviewng Editor and Reviewer #1, and the evaluation has been overseen by Volker Dötsch as the Senior Editor. The following individuals involved in the review of your submission have agreed to reveal their identity: Jyh-Yeuan Lee (Reviewer #2); Jean-Michel Jault (Reviewer #3).

Essential revisions:

1) We think it would be appropriate in the discussion to:

i. put this study in context with other approaches used to assess a synergetic effect of drugs and ATP to reach the OF state for P-gp (Futamata et al., 2020, J. Biol. Chem; Marcoux et al., 2013, PNAS).

ii. Clarify the limitations of the study, in particular, that the approach only has a read-out for the inward-facing state, half-life of antibody complex, and its inability to directly assess intermediate states and allosteric coupling between the NBDs and TMD regions.

2) Figure 1: It is unclear why in the presence of ABCG2 ({plus minus}GFP), the addition of Ko143 induces more fluorescence intensity. It looks like there is more 5D3 binding in the presence of Ko, like if this inhibitor could displace the equilibrium towards the IF state of ABCG2. Do you have other antibodies towards ABCG2 that will recognize it regardless of its conformation? Is the presence of other inhibitors or drugs would produce the same effect? How do the authors explain this result?

*Reviewer #1 (Recommendations for the authors):*

Overall, I found the paper was technically sound and integrated well with previous structural and biochemical analyses.

*Reviewer #2 (Recommendations for the authors):*

I commend your achievement in this study, especially the solid kinetic data. Thank you for the effort. The following are a few suggestions that I believe would make a stronger argument for your conclusion.

1. It is not clear to me that the data shows interdomain interaction beyond saying a global switch of IF and OF conformers. There have been a number of mutational studies done at the NBD-TMD interface, either canonical catalytic or disease mutations. By any chance, have the authors looked at the impact of those mutants?

2. Have you tried to swap the 5D3 epitope on ABCG2 with Pgp's UIC epitope, and you still reached the same conclusion?

3. This is a minor suggestion to improve Figure 5. The colocalization images are not clear to me on my PDF copy; thus, it would help to reformat the images to better present colocalization.

*Reviewer #3 (Recommendations for the authors):*

Line 96: when the authors speak about cavities 1 and 2, it would be useful to show them on the 3D structures of ABCG2. I think they could combine that with Figure 8 which would become the new figure 1 with different panels, some for the cavities and others for the 5D3 epitopes.

Figure 1: It is unclear why in the presence of ABCG2 ({plus minus}GFP), the addition of Ko143 induces more fluorescence intensity. It looks like there is more 5D3 binding in the presence of Ko, like if this inhibitor could displace the equilibrium towards the IF state of ABCG2. Do you have other antibodies towards ABCG2 that will recognize it regardless of its conformation? Is the presence of other inhibitors or drugs would produce the same effect? How do the authors explain this result?

Line 183. It is not so unexpected since, in the case of MsbA, APD/Mg/Vi was capable to switch a large proportion of transporter to the OF conformation (Moeller et al., 2015, Structure). This paper should be mentioned in the discussion.

Lines 216/217. 'Transported substrates increase the turnover rate of ATP hydrolysis in ABC

transporters'. In fact, it is found in many ABC transporters including eukaryotic multidrug transporter but this is not a general property of all ABC transporters.

Lines 277/278. '…probably by reducing the energy barrier of the above conformational changes'. You could mention here that this has been proposed for other multidrug ABC transporters when the basal ATPase is strongly stimulated by drugs (Orelle et al., 2022, Trends Microbiol).

Overall, the discussion part could be developed a little bit more and include two major publications that showed using different techniques a synergetic effect of drugs and ATP to reach the OF state for P-gp (Futamata et al., 2020, J. Biol. Chem; Marcoux et al., 2013, PNAS).

Lines 463/464, 'ABCG2 can transition from the IF to OF state, and this transition is not

the rate-limiting step of the whole catalytic cycle'. I don't think that this statement can be made from the data obtained. Clearly, the drugs seem to promote the transition from IF to OF, so affect the energetic barrier for this transition, yet this does not inform us of the rate-limiting step of the transport cycle. I suggest that the authors remove the last part of the sentence.

---

## [Author Response]

Essential revisions:1) We think it would be appropriate in the discussion to:i. put this study in context with other approaches used to assess a synergetic effect of drugs and ATP to reach the OF state for P-gp (Futamata et al., 2020, J. Biol. Chem; Marcoux et al., 2013, PNAS).

Futamata et al. analyzed nucleotide-induced conformational changes of ABCB1 using FRET. In agreement with the data presented in our manuscript, they find that ATP binding causes conformational changes leading to the outward-facing state, whereas the transported substrate verapamil does not induce a measurable FRET signal without the presence of MgATP. Marcoux et al. use an original approach to probe ABCB1 in a detergent micelle. Based on ion mobility MS analysis, the authors show that ABCB1 exists in an equilibrium between different states, readily interconverted by ligand binding. Both papers are now mentioned, to offer a broader context of our findings (please see lines 466-469 and 438439 in the amended manuscript).

ii. Clarify the limitations of the study, in particular, that the approach only has a read-out for the inward-facing state, half-life of antibody complex, and its inability to directly assess intermediate states and allosteric coupling between the NBDs and TMD regions.

We agree with the Reviewers that the antibody-shift assay has limitations, which should be clearly discussed, and therefore, we included a new paragraph in the Discussion (please see lines 402-409 in the revised manuscript).

We would like to make two comments here.

First, we believe that the half-life of the antibody complex does not limit our conclusions.

Second, the apparent ATP affinity determined in 5D3 reactivity assays (*K*_A_) (Figure 3 and Table 1) should be identical to the *K*_M_ of ATP for ATP-hydrolysis observed at the same (fixed) 5D3 concentration. Since 5D3 stabilizes the IF conformation of ABCG2, it should act as a mixed type inhibitor of ABCG2-mediated ATP-hydrolysis and substrate transport, by lowering the apparent *V_max_* and by increasing the apparent *K_M_* for ATP. Therefore, the absolute *K_A_* values were not used in our reasoning, only differences and changes were interpreted.

2) Figure 1: It is unclear why in the presence of ABCG2 ({plus minus}GFP), the addition of Ko143 induces more fluorescence intensity. It looks like there is more 5D3 binding in the presence of Ko, like if this inhibitor could displace the equilibrium towards the IF state of ABCG2.

Several inhibitors of ABCG2 enhance 5D3 mAb binding probably by shifting the equilibrium to the IF state of the transporter (*J Biol Chem,* 280, 421927.10.1074/jbc.M411338200; Eur J Pharm Sci (2012)45:101-109). Similar phenomenon has been described for UIC2 binding to ABCB1 earlier (e.g., Mechetner, E. B. et al. Proceedings of the National Academy of Sciences of the United States of America 94, 12908-12913 (1997); Goda, K. et al. The Journal of Pharmacology and Experimental Therapeutics 320, 81-88, doi:10.1124/jpet.106.110155 (2007)).

Note that Ko143 treatment does not affect the ABCG2-GFP fluorescence in the plasma membrane (Figure 5C), indicating that the Ko143-induced shift can be explained by a shift in the equilibrium towards the 5D3-reactive IF conformer, thereby increasing the number of transporters available for 5D3 binding. This is now explained in the text (please see lines 157-162).

Do you have other antibodies towards ABCG2 that will recognize it regardless of its conformation?

Intriguingly, (according to our knowledge) such an antibody (conformation-insensitive, recognizing an extracellular ABCG2 epitope) does not exist.

Is the presence of other inhibitors or drugs would produce the same effect? How do the authors explain this result?

As stated above, several compounds interacting with ABCG2 (inhibitors or substrates added at high (inhibitory) concentrations) result in an increase of 5D3-reactivity (see e.g., Telbisz et al. Eur J Pharm Sci (2012)45:101-109). A similar phenomenon was studied in great detail with ABCB1, based on the drug-induced shift of UIC2 binding (Mechetner, E. B. et al. Proceedings of the National Academy of Sciences of the United States of America **94**, 12908-12913 (1997); Goda, K. et al. The Journal of pharmacology and experimental therapeutics **320**, 81-88, doi:10.1124/jpet.106.110155 (2007).)

As also demonstrated in the case of ABCB1 (Mechetner, E. B. et al. Proceedings of the National Academy of Sciences of the United States of America **94**, 12908-12913 (1997); Goda, K. et al. The Journal of Pharmacology and Experimental Therapeutics **320**, 81-88, doi:10.1124/jpet.106.110155 (2007)), the likely explanation is that such compounds stabilize the 5D3-reactive (or UIC2-reactive) IF conformer by hindering the transition of the transporters to the OF state.

Reviewer #2 (Recommendations for the authors):I commend your achievement in this study, especially the solid kinetic data. Thank you for the effort. The following are a few suggestions that I believe would make a stronger argument for your conclusion.1. It is not clear to me that the data shows interdomain interaction beyond saying a global switch of IF and OF conformers. There have been a number of mutational studies done at the NBD-TMD interface, either canonical catalytic or disease mutations. By any chance, have the authors looked at the impact of those mutants?

We agree that our data reflect a global switch between IF and OF conformers. In the current work we have not studied the impact of mutations introduced into the NBD-TMD interface on the conformation changes of the protein.

2. Have you tried to swap the 5D3 epitope on ABCG2 with Pgp's UIC epitope, and you still reached the same conclusion?

Very interesting thought, but in our humble opinion, this would be more than a difficult task and most likely cannot be achieved. Both epitopes are composite epitopes that are transiently formed during the catalytic cycle of the transporters. The UIC2 epitope is formed by the 1^st^, 3^rd^ and 4^th^ extracellular loops of the human ABCB1 (Zhou, Y. at al. (1999). *Archives of Biochemistry and Biophysics*
**367**, 74-80 Vahedi, S. et al. 2018. *Scientific Reports*
**8**, 12716; Alam, A. et al. (2019) *Science*
**363**, 753-756), while the 5D3 epitope is formed mainly by the 3^rd^ extracellular loop connecting TM5c and TM6a in the ABCG2 protomers (Taylor NMI et al. *Nature,* 546**,** 504-509.10.1038/nature22345). The ABCG2 dimer collectively forms two epitopes for 5D3 Fab binding. In contrast, only one UIC2 Fab can bind to ABCB1.

Even in case the amino acids in these loops could in some way be switched between the transporters, this would not lead to the transfer of the epitopes between transporters, because the folds of the extracellular loops are very different, as are the folds of the transmembrane helices.

3. This is a minor suggestion to improve Figure 5. The colocalization images are not clear to me on my PDF copy; thus, it would help to reformat the images to better present colocalization.

We replaced the images with high-quality and high-resolution tif files.

Reviewer #3 (Recommendations for the authors):Line 96: when the authors speak about cavities 1 and 2, it would be useful to show them on the 3D structures of ABCG2. I think they could combine that with Figure 8 which would become the new figure 1 with different panels, some for the cavities and others for the 5D3 epitopes.

We added two panels to Figure 8 showing the cavities in the IF and OF conformations (and relocated this Figure into the Introduction part as a new Figure 1.)

Line 183. It is not so unexpected since, in the case of MsbA, APD/Mg/Vi was capable to switch a large proportion of transporter to the OF conformation (Moeller et al., 2015, Structure). This paper should be mentioned in the discussion.

We agree – thank you for the suggestion (please see lines 197-199 in the revised manuscript).

Lines 216/217. 'Transported substrates increase the turnover rate of ATP hydrolysis in ABC transporters'. In fact, it is found in many ABC transporters including eukaryotic multidrug transporter but this is not a general property of all ABC transporters.

We have modified this sentence according to the Reviewers’ suggestion (please see lines 231/232).

Lines 277/278. '…probably by reducing the energy barrier of the above conformational changes'. You could mention here that this has been proposed for other multidrug ABC transporters when the basal ATPase is strongly stimulated by drugs (Orelle et al., 2022, Trends Microbiol).

Thank you for this suggestion, we have inserted the reference (please see lines 292/293).

Lines 463/464, 'ABCG2 can transition from the IF to OF state, and this transition is not the rate-limiting step of the whole catalytic cycle'. I don't think that this statement can be made from the data obtained. Clearly, the drugs seem to promote the transition from IF to OF, so affect the energetic barrier for this transition, yet this does not inform us of the rate-limiting step of the transport cycle. I suggest that the authors remove the last part of the sentence.

Thank you for this comment. We have reformulated the corresponding section to clarify our reasoning (please see lines 474-508).